# Periostin Circulating Levels and Genetic Variants in Patients with Non-Alcoholic Fatty Liver Disease

**DOI:** 10.3390/diagnostics10121003

**Published:** 2020-11-25

**Authors:** Carlo Smirne, Violante Mulas, Matteo Nazzareno Barbaglia, Venkata Ramana Mallela, Rosalba Minisini, Nadia Barizzone, Michela Emma Burlone, Mario Pirisi, Elena Grossini

**Affiliations:** 1Department of Translational Medicine, Università del Piemonte Orientale, via Solaroli, 17, 28100 Novara, Italy; viola.mulas@gmail.com (V.M.); matteo.barbaglia89@gmail.com (M.N.B.); vramana6565@gmail.com (V.R.M.); rosalba.minisini@med.uniupo.it (R.M.); michela.burlone@med.uniupo.it (M.E.B.); mario.pirisi@med.uniupo.it (M.P.); elena.grossini@med.uniupo.it (E.G.); 2Department of Health Sciences, Università’ del Piemonte Orientale, via Solaroli, 17, 28100 Novara, Italy; nadia.barizzone@med.uniupo.it

**Keywords:** periostin, biomarker, non-alcoholic fatty liver disease, non-alcoholic steatohepatitis, hepatocellular carcinoma, liver steatosis, liver fibrosis, single nucleotide polymorphism, metabolic syndrome, extracellular matrix

## Abstract

Circulating periostin has been suggested as a possible biomarker in non-alcoholic fatty liver disease (NAFLD) in Asian studies. In the present study, we aimed to test its still controversial relevance in a Caucasian population. In patients with histologically-proven NAFLD (N. = 74; 10 with hepatocellular carcinoma, HCC) plasma periostin concentrations were analyzed. POSTN haplotype analysis was based on rs9603226, rs3829365, and rs1029728. Hepatitis C patients (N. = 81, 7 HCC) and healthy subjects (N. = 27) were used as controls. The median plasma periostin concentration was 11.6 ng/mL without differences amongst groups; it was not influenced by age, liver fibrosis or steatosis. However, possession of haplotype two (rs9603226 = G, rs3829365 = C, rs1028728 = A) was associated with lower circulating periostin compared to other haplotypes. Moreover, periostin was higher in HCC patients. At multivariate analysis, HCC remained the only predictor of high periostin. In conclusion, plasma periostin concentrations in Caucasians NAFLD patients are not influenced by the degree of liver disease, but are significantly higher in HCC. Genetically-determined differences may account for some of the variability. These data suggest extreme caution in predicting a possible future role of periostin antagonists as a rational therapeutic alternative for NAFLD, but show a potential periostin role in the management of NAFLD-associated HCC.

## 1. Introduction

Non-alcoholic fatty liver disease (NAFLD), the major cause of chronic liver disease worldwide, includes a spectrum of chronic diseases ranging from simple steatosis (SS) to non-alcoholic steatohepatitis (NASH). Some patients with NASH are likely to develop into cirrhosis and even hepatocellular carcinoma (HCC) [1]. The mechanisms that lead to NAFLD development are complex and multifactorial, and have not yet been fully clarified. They range from environmental factors to genetic variants resulting in a disturbed lipid homeostasis and an excessive accumulation of triglycerides (TG) and other lipid species in hepatocytes [2,3,4,5].

During the last years, a major focus in the deeper understanding of pathogenesis of this condition allowed the discovery of several novel mediators and promising targets. Amongst the factors that have a potential role in driving aberrant accumulation of TGs in the liver, emerging evidence indicates that the dysfunction of periostin (PN) expression plays a prominent action. PN, also known as osteoblast-specific factor 2 (OSF-2), is a 90 kDa multifunctional extracellular matrix (ECM) protein, coded by periostin (*POSTN*) gene and mainly secreted by osteoblasts [6]. It has pleiotropic activities far beyond simple bone remodeling; as a matter of fact, it is also involved—amongst others—in the pathophysiology of arthritis, atherosclerosis, and inflammatory diseases [7]. Actually, one of the target organs in which PN has been shown to play a crucial role is the liver, where it can modulate the cell fate determination and proliferation, inflammatory responses, ECM remodeling, even tumorigenesis [8,9,10,11,12]. In this respect, the strongest evidence so far concerns its pivotal role in the onset of metabolic disease (such as obesity and glucose or lipid disorders) by suppression of fatty acid oxidation in the liver [13]. As a matter of fact, in obese mice the overexpression of PN in the liver was shown to induce hepatic steatosis and hypertriglyceridemia through the downregulation of peroxisome proliferator-activated receptor (PPAR)-α, which activates the fatty acid oxidation in mitochondria and peroxisomes [14]. Conversely, the genetic knockout of PN significantly improved those conditions [13] and was able to protect mice against dietary-induced NAFLD [15]. In detail, PN could exert its protective effects against hypertriglyceridemia and liver steatosis through the interaction with the subtype α6β4 of the integrins family and the subsequent activation of an intracellular pathway involving Ras-related C3 botulinum toxin substrate (Rac) 1 and c-JUN. Those events would lead to the inhibition of the PPAR-α promoter, RAR-related orphan receptor (ROR) α, and to the downregulation of PPAR-α itself [13].

Considering human studies, the relationship between PN and hepatic steatosis, which is the basis of the present study, was first proposed by Lu et al. [13], showing an upregulation of hepatic PN expression in NAFLD patients, with a good correlation with hepatic TG content. Moreover, serum PN levels were found to be increased in the same subjects, although without a significant correlation with hepatic TGs. Additionally, in the studies from Zhu et al. and Yang et al. higher plasma PN concentrations were observed in NAFLD patients in comparison to healthy controls, although without significant differences between the ultrasound degrees of NAFLD [16,17]. In addition, circulating PN was strongly associated with lipid metabolism, chronic inflammation and insulin resistance [17,18], and the serum and liver tissue levels of PN were closely related to the decline in liver function and to the pathological stage in NAFLD patients [8,17]. It is important to note, that plasma PN analysis—performed through the same enzyme linked immunosorbent assay (ELISA) kit of the research from Zhu et al. on Chinese subjects—failed to find any association between circulating PN and hepatic steatosis on a small but histologically-confirmed casuistry of NAFLD European patients, while there was a statistically significant trend for PN level decrease in more severe fibrosis stages [19].

Besides steatosis, PN has been demonstrated to mediate also hepatic inflammation and fibrosis, probably through both direct and indirect mechanisms related to the concomitant release of pro-inflammatory and pro-fibrotic factors and inhibition of adiponectin, an insulin-sensitizing and anti-inflammatory adipokine. Despite this experimental evidence, there are currently limited data for PN in human NAFLD evolved to NASH or hepatic fibrosis/cirrhosis (which could be expected to present higher PN levels than SS patients) [8,20].

The same lack of data applies to another possible NAFLD complication, i.e., HCC. To the best of our knowledge, in fact, no specific evidence already exists for a PN role. However, it is reasonable to assume that, at least in principle, many of the considerations that have been made for HCCs in general are valid. Regarding this issue, an important PN expression has been shown in the tumor stroma, associated with a greater co-expression of vascular endothelial growth factor (VEGF), and a reduced overall, and disease-free, survival [10,21]. Additionally, circulating levels of PN were demonstrated to be higher in HCC patients compared to healthy controls [22].

Coming back to NAFLD, taking into account the aforementioned discordant results found by researchers in circulating PN levels, and assuming that they were at least in part related to ethnic differences, it is interesting to note that the *POSTN* gene, like many others, has some known genetic variants. In particular, four major and two minor haplotypes could be identified (initially, in the Japanese population). The former ones were mainly determined by three tag single nucleotide polymorphisms (SNPs), which were located at intron 66 bp upstream of exon 21 (rs9603226, alleles G>A), 5′ untranslated region (rs3829365, alleles G>C), and at promoter region (rs1028728, alleles A>T) [23]. The four major *POSTN* haplotypes obtained, respectively, from these three SNPs were: AGA (haplotype one), GCA (haplotype two), GGT (haplotype three), and GGA (haplotype four). These SNPs were then studied in different human disorders, mainly in the cardiology and pulmonology fields, and the results seem to suggest a possible role of PN and of its allelic variants in identifying patients at increased risk of disease progression [24,25]. However, to the best of our knowledge, no specific evidence has been yet provided for liver diseases, including NAFLD.

Based on these premises, in the present study we aimed to provide implications and mechanisms of PN in the pathogenesis of NAFLD, confirming (or denying) the possible association between circulating PN and this specific disease (in all its stages including HCC), and evaluating—for the first time in hepatology—the role of genetic factors as possible contributors to the observed variability in the plasma concentration of this protein.

## 2. Materials and Methods

### 2.1. Patients

For this retrospective cross-sectional study over a period of nine years (2008–2016), patients with NAFLD confirmed diagnosis were recruited, including some subjects complicated with HCC. CHC subjects, again including some with HCC, and healthy volunteers were used as control groups. The research was conducted at the Liver Clinic of Novara University Hospital, which is a large tertiary center (serving a population of about 1,000,000). The study protocol was carried out in accordance with the declaration of Helsinki. Written informed consent for the storage of biological material (blood) and clinical, histological and laboratory data processing was obtained from all the participants.

#### 2.1.1. NAFLD Patients

Inclusion criteria were: (1) adult age; (2) clinical diagnosis of NAFLD, confirmed by liver biopsy in the case of disease not complicated by cancer, and by instrumental and/or histological means when an HCC was present; (3) availability of a blood sample with authorization to carry out the genetic investigation (*POSTN* genotyping); (4) availability of a fasting blood sample collected no more than seven days after the liver biopsy with authorization to perform PN ELISA quantification. Exclusion criteria were as follows: (1) hepatitis C virus (HCV) positive serology; (2) Australia antigen (HBsAg) positivity; (3) excessive alcohol use at the biopsy time and/or in the previous 6 months (more than one drink per day for women and two drinks per day for men, and more than one drink per day for individuals over the age of 65) [26,27]; (4) other concomitant causes of liver disease.

Based on these criteria, of the 107 subjects with NAFLD that were initially identified at different stages (i.e., fatty liver alone, non-alcoholic steatohepatitis (NASH), cirrhosis, and liver cancer) 33 were then excluded: 3 for HBsAg positivity, 5 for habitual excess alcohol consumption, and 25 for non-availability of whole blood and/or plasma samples. The tested population was therefore of 74 Caucasian subjects, 10 of whom with HCC at the time of diagnosis. All subjects underwent a liver biopsy, except for 5 HCC subjects who were diagnosed on the basis of radiological imaging, as required by the 2011 EASL-EORTC (European Association for the Study of the Liver, EASL; the European Organization for Research and Treatment of Cancer, EORTC) guidelines [28]. HCC staging was according to Barcelona clinic liver cancer (BCLC) guidelines [28,29]. Non-tumor patients had their grading and staging histological assessment according to Kleiner NAFLD Activity Score [30]. Moreover, a non-invasive assessment of liver fibrosis was performed in the same patients both through transient elastography (TE) (FibroScan^®^, Echosens, Paris, France), which measures liver stiffness (LS), and NAFLD fibrosis score (NFS) [31]. Only TE measurements taken no more than six months after liver biopsy were taken into account; furthermore, examinations were considered valid as long as they were technically feasible and satisfied the known quality criteria as previously described [32]. Whenever the correct probe was available, TE device was used to assess, at the same time, LS, which as previously reported is related to liver fibrosis, and the CAP, indicative of the degree of liver steatosis [33].

A systematic data collection was carried out by review of all medical records at baseline (time of liver biopsy or HCC radiological diagnosis). Age, gender, race, height, and weight were documented, and BMI was calculated as weight (in kilograms) divided by the square of the height (in meters). History of diabetes mellitus, hyperglycemia, or hyperlipidemia was ascertained. Laboratory data included, amongst others, the following: blood count, serum AST, ALT, GGT, alkaline phosphatase, total bilirubin, albumin, total protein, creatinine, prothrombin time, international normalized ratio, fasting glucose, fasting insulin, and lipid panel. Liver biopsy, TE and blood tests were performed within 3 months in more than 80% of subjects, and between 3 and 6 months in the remaining cases.

#### 2.1.2. HCV Control Patients

A control population comprising age- and sex-matched Caucasian CHC patients was studied. Inclusion and exclusion criteria were the same as for NAFLD patients (except for inclusion criterion # 2 and exclusion criterion # 1). Of the 97 subjects initially identified, 7 were excluded for habitual excess alcohol consumption, and 9 for non-availability of whole blood and/or plasma samples, resulting in 81 patients included in the study. This population included 7 subjects with HCC at the time of diagnosis: two underwent a liver biopsy, and five were diagnosed on the basis of radiological imaging and staged similarly to NAFLD patients. CHC histological evaluation of non-tumor patients included grading and staging assessment using the Ishak classification [34], and fat percentage quantification. Subjects underwent TE for non-invasive assessment of liver fibrosis, when technically feasible, with the same considerations for validity and CAP measurement as for NAFLD patients.

The above considerations for the data collection described for NAFLD subjects also apply in this context, but with the addition of HCV genotype and viremia at the time of biopsy. HCV RNA was detected by COBAS AMPLICOR HCV Test, v2.0 (Roche Molecular Systems, Pleasanton, CA, USA).

#### 2.1.3. Healthy Controls

Twenty-seven Caucasian subjects in apparent good health and between 24 and 54 years old were studied. They were not age- and sex-matched with the patients, but selected according to the following criteria: normal weight (18.5 ≤ BMI < 25 kg/m^2^), no medical history of liver disease, asthma, diabetes, allergic syndromes, and/or high risk alcohol consumption as reported above. All underwent TE liver stiffness measurement, which was less than 7.0 kPa in all subjects, allowing to exclude the presence of advanced liver fibrosis. CAP was also measured in all, and was indicative of the absence of significant steatosis [33].

### 2.2. POSTN Genetic Studies

Genomic DNA was extracted from whole blood or buffy coat, using a commercial kit (Invitrogen, Carlsbad, CA, USA), according to the manufacturer’s instructions. DNA was then amplified by polymerase chain reaction (PCR). We analyzed three different polymorphism of *POSTN* gene (rs9603226, rs3829365 and rs1028728) using restriction fragment length polymorphism-PCR. The PCR primer sequences used for *POSTN* amplification were as follows: rs9603226 forward: 5′-ATGAATTTGGTGACCTTGGTG-3′, reverse: 5′-CAATCTATTGTTCATTTCCATACC-3′; rs3829365 forward: 5′-TTCAGGTTGATGCAGTGTTCC-3′, reverse: 5′-CCGACCCCTGATACGACT-3′; rs1028728 forward: 5′-GCAGCCAATATTGGAAGCAAG-3′, reverse: 5′-GGATGGTGTGCAGCTTGTTTATTC-3′. The PCR products of rs9603226, rs3829365 and rs1028728 were digested using the enzymes AhdI, Eam1104I (Life Technologies, Thermo Fisher Scientific, Carlsbad, CA, USA) and AgsI (SibEnzyme Ltd., Novosibirsk, Russia), respectively. All samples were amplified twice; when discordant, they were run a third time. The lab technicians performing the genetic studies were blinded about the case/control state of the subjects enrolled. As an additional check, some samples were sequenced by GATC Biotech AG (European Custom Sequencing Service, Cologne, Germany). The haplotypes were obtained using the program Beagle 4.1 [34,35]. For each patient, two haplotypes were assigned, based on the SNP association, as shown in Table 1. The haplotype analysis was performed using the software Haploview (Broad Institute of MIT and Harvard, Cambridge, MA, USA). Appendix A
Table A1 reports the diplotypes obtained from haplotype combinations and found in the studied population.

### 2.3. Plasma Periostin Concentration Dosage

Plasma samples were obtained from the recruited patients and healthy controls, and stored at −80 °C. The determination of plasma PN concentration (diluted 1:25) was performed by ELISA immunoassay on a 96-well plate (Human Periostin/OSF-2 DuoSet ELISA, R&D Systems Inc., Minneapolis, MN, USA) according to manufacturer’s instructions. Each sample was analyzed in duplicate and compared with a standard curve obtained by using different PN concentrations. The reading was performed through a spectrometer at 450 nm (Victor X4 Perkin Elmer, Milan, Italy).

### 2.4. Statistical Analysis

The statistical analysis was conducted with the help of Stata statistical software, version 13.1 (StataCorp LP, College Station, TX, USA). The test used to verify the normality of the data was that of Shapiro–Wilk, which confirmed that, for almost all of the continuous variables and as expected for the vast majority of biological parameters (in both health and disease conditions), the data distribution deviated significantly from the normal one. The centrality and data dispersion measures for the continuous variables were therefore median and IQR. The categorical variables were presented as frequencies (percentage of the total). For the continuous variables, the differences between the groups were analyzed with the Mann–Whitney (in the case of two independent groups) and Kruskal–Wallis (in the case of more than two independent groups) tests. The association between the categorical variables was explored using Fisher’s exact test or Pearson’s chi-square test, as appropriate. An extension of the Wilcoxon rank-sum test as proposed by Cuzick et al. was used to handle the situations in which a variable was measured for individuals in three or more (ordered) groups and a non-parametric test for trend across these groups was desired [36]. The strength and direction of the monotonic relationship between two ordinal variables was measured by Spearman’s Rho correlation coefficient. The concordance between invasive (histopathological examination) and non-invasive (NFS, TE) measures of liver fibrosis was assessed by calculating Cohen’s kappa index. The differences in a dependent variable (e.g., plasma PN values) among groups, taking into account its variability explained by one or more covariates, were tested through the analysis of covariance (ANCOVA). Finally, a set of variables (*POSTN* diplotype, age, sex, HCC, and platelet count) was analyzed in a maximum-likelihood logit regression model as a predictor of high plasma PN values (defined by belonging to the last quintile of the study population). For all the tests used, the value chosen to indicate the statistical significance threshold was 0.05 (two tailed).

## 3. Results

### 3.1. Patients and Healthy Controls

As extensively described in the Materials and Methods section, the study was centered on 74 patients with confirmed diagnosis of NAFLD. Hepatitis C virus (HCV) infected patients (N. = 81) and healthy individuals (N. = 27) were used as controls. Thus, the total sample consisted of 182 subjects (155 patients with chronic liver diseases and 27 healthy controls). NAFLD and HCV groups included a proportion of subjects affected by HCC, 10 (two stage zero, two stage A, five stage B, one stage C) and seven (two stage zero, two stage A, two stage B, one stage D), respectively. The main characteristics of the patients and the healthy controls included in the study are shown in Table 2; Table 3, respectively. The distribution of NAFLD patients according to the Kleiner classification system is detailed in Appendix A
Table A2.

Patients’ biochemical parameters were obtained after at least 8 h of fasting. As can be seen, these were subjects that can be considered reasonably representative of the type of patients who typically accesses a hepatology unit for the diagnosis and treatment of the respective pathologies. As far demographic characteristics are concerned, the two groups were similar, both for median age and the sex ratio. Metabolic parameters such as glycaemia, total cholesterol and triglycerides were significantly more elevated in the NAFLD group. The same applies for the frequencies of obesity, diabetes or prediabetes. Specularly, it appears consistent with belonging to the HCV group, that the levels of aspartate aminotransferase (AST) and alanine aminotransferase (ALT) were higher in this subpopulation, which is more commonly characterized by increased cytonecrotic damage.

Considering only non-cancer patients (N. = 138), the cases of advanced fibrosis (defined as a stage ≥3 or ≥4 according to Kleiner or Ishak classifications, respectively) (N. = 48) showed similar proportions between the two groups, although with some excess prevalence in the NAFLD group. The same considerations applied for the liver stiffness. Concerning the frequency of HCV patients with advanced fibrosis, it is consistent with the estimates of disease stage distribution in Italy (and, more generally, in Europe) before the spread of direct antiviral agents (DAA). Focusing only on patients with established cirrhosis (defined as a stage four or six according to Kleiner or Ishak classifications, respectively) (N. = 16), they were more prevalent in NAFLD than in HCV group (19% vs. 5%, respectively, *p* = 0.01); all subjects had Child-Pugh score A, and the median score (interquartile range, IQR) of the Model for End-Stage Liver Disease (MELD) was eight (seven to eight). With regard to hepatic steatosis, the NAFLD group (N. = 64), as expected, had significantly higher controlled attenuation parameter (CAP) values than the chronic hepatitis C (CHC) subset (N. = 74) (*p* = 0.003). Additionally, histological steatosis, quantified as the percentage of steatotic hepatocytes (up to 33%, between 33 and 66%, and more than 66%), was significantly more severe in NAFLD patients compared to HCV subjects: 21 (33%), 30 (47%), and 13 (20%) vs. 67 (91%), 4 (5%), and 3 (4%) patients, respectively (*p* < 0.001). In particular, the NAFLD group had a higher prevalence of severe (i.e., more than 66%) steatosis (*p* = 0.006).

### 3.2. Plasma Periostin Concentrations

#### 3.2.1. Periostin Concentrations and Disease Etiology

The median plasma PN value (IQR) in the study population (N. = 182) was 11.6 ng/mL (8.7–13.3). In particular, it was 10.9 ng/mL (7.7–14.6) in the NAFLD group (N. = 74), 12.0 ng/mL (9.8–16.5) in the HCV group (N. = 81), and 12.0 ng/mL (9.1–16.3) in the healthy subject group (N. = 27), respectively, with a trend toward statistical significant differences amongst groups (*p* = 0.06). Figure 1 shows the scatterplot of the distribution of plasma PN values among the different disease etiologies. The red symbols identify cases of HCC; it can be observed that these subjects had PN concentrations that were always equal to or higher than the median of the respective groups -with the exception of one single NAFLD case (*p* = 0.006 and 0.004 for NAFLD and HCV patients, respectively) (also see Table 4).

Considering all patients (N. = 155), differences among sexes with regard to PN levels did not reach statistical significance, either considering them as a whole (median in males: 11.9 ng/mL, IQR 8.2–16.8; median in females 11.1 ng/mL, IQR 8.5–14.8, *p* = 0.196), or separately analyzing the two subgroups, i.e., HCV (median in males: 12.9 ng/mL, IQR 10.7–17.4; median in females 11.3 ng/mL, IQR 9.2–16.0, *p* = 0.275) and NAFLD (median in males: 11.8 ng/mL, IQR 7.8–16.1; median in females 10.5 ng/mL, IQR 7.1–13.3, *p* = 0.418).

#### 3.2.2. Periostin Concentrations and Hepatic Steatosis Degree

Additionally, no differences could be found between circulating PN levels and the degree of hepatic steatosis in cancer-free patients (N. = 138), quantified as the following percentages of steatotic hepatocytes: up to 33%, between 33 and 66%, and more than 66% (for the number of individual groups, see Section 2.1). For what concerns NAFLD group (N. = 64), median values were 10.4 (8.0–13.0), 11.0 (7.5–14.6), and 8.1 (7.0–13.8) ng/mL, respectively (*p* = 0.553). Hepatitis C patients (N. = 74) had the following concentrations: 11.2 (8.6–15.3), 13.1 (9.2–16.7), and 11.5 (10.5–15.5) ng/mL, respectively (*p* = 0.763). Similarly, no correlation was observed between plasma PN levels and the three degrees of steatosis both in NAFLD (Spearman Rho = −0.089, *p* = 0.488) and HCV (Spearman Rho = 0.139, *p* = 0.236) subjects. Finally, no correlation was found between PN and CAP values in the NAFLD, HCV and healthy control subjects (Spearman Rho = 0.084, −0.303, and −0.252, respectively; *p* = 0.688, 0.151, and 0.215, respectively).

#### 3.2.3. Periostin Concentrations and Hepatic Fibrosis

Table 3 panel a shows the values of PN soluble concentrations according to the severity of the disease in the various study groups, including healthy controls: significant differences were found within total and patient populations. Table 3 panel b describes the comparisons of PN plasmatic levels within the various fibrosis stages in non-HCC individuals (again, including healthy controls): no significant differences could be found. Table 3 panel c shows the comparison of PN plasmatic levels in HCC vs. non-HCC patients (NAFLD and HCV) divided according to the various fibrosis stages: HCC subjects showed always significantly higher PN values compared to any stages of fibrosis in their respective groups.

For what concerns non-HCC patients, no correlation was observed between plasma PN levels and the three stages of fibrosis both in NAFLD (N. = 64) (Spearman Rho = 0.079, *p* = 0.534) and HCV (N. = 74) (Spearman Rho = 0.139, *p* = 0.237) subjects.

As regards HCC patients, no significant differences were found between NAFLD (N. = 10) and HCV (N. = 7) subjects (*p* = 0.187) (Figure 2 panel a). Additionally, no differences were found amongst various BCLC HCC stages (*p* = 0.27) (Figure 2 panel b), although there was a trend for increased PN levels in more advanced stages: 14.9 (11.9–17.6) vs. 18.3 (14.4–38.8) ng/mL in stages 0–A vs. B–C–D, respectively (Figure 2 panel c) (*p* = 0.09).

Always with regard to HCC patients, a further analysis was conducted on serum alpha-fetoprotein, which was dosed, as clinical practice, in all HCC patients at the same time of PN sampling. The concentrations were as follows in the different BCLC stages: 31.9 (8.9–79.8), 18.7 (2.6–672.0), 19.9 (4.0–601.0), 276.3 (276.3–276.3), and 9.2 (9.2–9.2) ng/mL, for stages 0, A, B, C and D, respectively. No significant differences were found in alpha-fetoprotein concentrations within the total HCC population (*p* = 0.834) or when HCC stages were grouped as 0–A (23.1 (4.5–79.8) ng/mL) and B–C–D (19.9 (4.7–438.6) ng/mL) (*p* = 0.704), as performed for the analysis of PN. Finally, a correlation was observed between circulating PN and alpha-fetoprotein levels (Spearman Rho = −0.487, *p* = 0.047).

#### 3.2.4. Periostin Concentrations and Main Demographic and Clinical Variables

When PN concentrations were analysed according to the main clinical variables of the entire study population (N. = 182), no statistically significant associations were observed among the various age groups (<40, 40–60, >60 years; *p* = 0.356), sex (*p* = 0.203), and body mass index (BMI) (*p* = 0.521) categorized, according to the cutoffs indicated by the World Health Organization [37]. PN levels had a trend to be inversely related to BMI (Spearman Rho = −0.134, *p* = 0.086), and did not vary in relation to age (Spearman Rho = 0.064, *p* = 0.392).

A similar analysis was conducted within the patient population (NAFLD and HCV groups, N. = 155). considering glucose metabolism status (normal glucose tolerance, prediabetes, diabetes) and some main laboratory variables: ALT (cutoffs: <40, 40–99, ≥100 IU/L), creatinine (cutoffs: <1.0, 1.0–1.4, ≥1.5 mg/dL), total cholesterol (cutoffs: <200, 200–240, ≥240 mg/dL), triglycerides (cutoffs: <80, 80–149, ≥150 mg/dL) and platelets (cutoffs: ≥150, 100–149, <100 × 109/L). No statistically significant associations could be identified for the first five studied parameters (*p* = 0.749, 0.810, 0.715, 0.543, and 0.861, respectively). Instead with regard to the aforementioned platelet cutoffs—commonly used in clinical practice to indicate a mild or a moderate thrombocytopenia—a progressive significant decrease in the plasma levels of PN was observed (*p* = 0.007): 15.2 (11.9–18.3) ng/mL (N. = 115), 13.1 (10.5–17.7) ng/mL (N. = 27), and 11.0 (7.99–16.3) ng/mL (N. = 13) for platelets ≥150, 100–149, <100 × 109/L, respectively. However, no correlation was found between the individual PN and platelet levels (Spearman’s Rho = 0.102, *p* = 0.205). Finally, when adjusted for the concentrations of serum glucose and serum TGs, plasma PN levels (after logarithmic transformation) were significantly higher in the HCV group (N. = 74) than in the NAFLD group (N. = 64) (geometric mean ± standard deviation (SD): 12.9 ± 1.6 vs. 10.2 ± 2.5 ng/mL, respectively; *p* < 0.001).

### 3.3. POSTN Genotyping

#### 3.3.1. Genotypic Frequencies

The studied population was in Hardy–Weinberg equilibrium for all three (rs9603226, rs3829365 and rs1028728) *POSTN* gene SNPs (*p* = 0.135, 0.224, and 0.580, respectively). Table 5 reports the specific locus of each SNP (obtained from the website www.ncbi.nlm.nih.gov/snp) and shows the distributions of the polymorphisms in the entire study population and in each individual group (NAFLD, HCV, controls). The most frequent genotypes for each SNP, both globally and in single groups, were G/G, C/C and A/A, respectively (thus being considered as homozygous wild types). Specularly, the less common variants were A/A, G/G and T/T, respectively.

The *POSTN* haplotype full sequences are shown in Table 1 belonging to the Materials and Methods section.

#### 3.3.2. Periostin and *POSTN* Gene Polymorphisms

Plasma PN concentrations were evaluated as a function of the three *POSTN* gene polymorphisms. Table 6 shows the median concentrations (IQR) of PN, both in the total studied population and in the three groups. No statistically significant correlations were observed between the genotype of the single polymorphisms and PN circulating levels.

Dominant and recessive genetic models were then calculated by referring to the allele frequency, considering the most frequent allele as the major one (M), and the least frequent allele as the minor one (m) [38]. Table 7 panel a shows plasma PN concentrations as a function of a dominant model (thus comparing MM versus Mm + mm): no statistically significant differences in PN levels with regard to the aforementioned variant alleles in the whole patient population were reported (N. = 155). The same also happened when applying a recessive model, and comparing MM + Mm versus mm (Table 7 panel b).

#### 3.3.3. Periostin and *POSTN* Gene Diplotypes

We analyzed the frequencies of the *POSTN* gene haplotypes in the entire study population and in the individual groups. In all cases (in the total studied population, NAFLD patients, HCV subjects, and healthy controls, respectively) haplotype two (allelic frequencies of 0.596, 0.602, 0.574, 0.648) was significantly more prevalent than haplotype one (not detected), haplotype three (frequencies of 0.003, 0, 0.006, 0), haplotype four (frequencies of 0.025, 0.020, 0.037, 0), haplotype five (frequencies of 0.137, 0.142, 0.136, 0.130), and haplotype six (frequencies of 0.239, 0.236, 0.247, 0.222).

An analysis was then conducted of circulating PN values as a function of the diplotypes (i.e., matched pairs of haplotypes on homologous chromosomes), considering the most frequent haplotype (number two, as previously mentioned) as the major one. Diplotype 2/* (i.e the combination of haplotype two and a haplotype other than 2), was the most frequent, both in the total study population and in the NAFLD, HCV and healthy controls groups (Table 8). In no case it was possible to detect any statistically significant difference amongst the groups in plasma PN levels. However, when analyzing patients carrying haplotype two (i.e., diplotypes 2/2 and 2/*) against other diplotypes, the former ones had more frequently plasma PN levels that were lower than the last quintile of the total studied population (130/156 vs. 16/26, respectively; *p* = 0.01).

Significant differences in steatosis grading and fibrosis staging were found between different diplotypes (either applying dominant and recessive models): *p* was <0.001 in all cases, both in the total studied population and in the NAFLD group.

### 3.4. Multivariate Analysis of Factors Associated with High Periostin

Multivariate analysis among those variables with *p* < 0.10 at univariate analysis was performed to identify possible predictors of high plasma levels of PN (i.e., those included in the last quintile) in the whole patient population (N. = 155). The set of covariates included *POSTN* diplotype (categorized as 2/2 plus 2/* versus */*), age, sex, HCC, and platelet count. The significant predictor variables found were *POSTN* diplotype and HCC (Table 9 panel a). The same results were confirmed when the analysis was restricted to the two single patient groups (i.e., NAFLD and HCV, Table 9 panels b and c, respectively), with the exception of *POSTN* diplotype in NAFLD subjects.

## 4. Discussion

This is the first study aiming to examine PN plasma levels and polymorphisms in NAFLD and other different pathologic conditions (namely CHC and HCC), and their possible relations with the progression of liver disease. The results shown evidence a poor association between circulating PN levels and the etiology of liver disorders. In addition, no correlation was shown with the severity of NAFLD or its fibrotic evolution. Interestingly, however, the results obtained showed that plasma PN was influenced by genotype and HCC.

PN is a known important factor in metabolic disease pathogenesis, and is directly and indirectly involved in hepatic steatosis and hypertriglyceridemia, which are key determinants of NAFLD. The evidence coming from the first studies which demonstrated the important role played by the protein in cellular and animal models [13,14,15] was then confirmed in human research, at least in Asian subjects [8,17,18]. Obviously, NAFLD pathogenesis cannot be reduced to fat accumulation alone. For instance, chronic inflammation should never be omitted, being a known contributor to the progression to NASH [39]. In this respect, many studies have suggested a close relationship between PN and liver fibrosis. In particular, hepatic inflammation and fibrosis were significantly inhibited in PN-knockout mice, mainly through a transforming growth factor (TGF)-β1 and TGF-β2 dependent mechanism, revealing a potential role of PN in NASH [9,15]. More recently, researchers have further conducted discussions on the intrinsic mechanisms of PN-induced hepatic fibrosis: collectively, PN has been shown to play an important role in activated hepatic stellate cells, aggravating the accumulation of fiber and matrix, ultimately inducing liver fibrosis [40,41,42]. As regards these issues, it should be emphasized that PN belongs to the so called “liver matrisome”, which identifies the whole of ECM and non-fibrillar proteins that interact or are structurally affiliated with the ECM, and are involved in the regulation of tissue homeostasis and organ function. In the liver, maladaptative changes of matrisome in response to any stress or injury—including liver inflammation and damage caused by a buildup of fat in the liver, the so-called NASH—can eventually lead to hepatic fibrosis [43]. The role of PN as a profibrotic agent through the liver matrisome modulation could be related to its direct actions on key molecules or proteins of ECM or on enzymes involved in the collagen stabilization and matrix integrity and elasticity, such as lysyl oxidases (LOX) and LOX like enzymes (LOXL). Hence, PN was shown to cause collagen expression in HSCs via integrin αvβ3 and the activation of the phosphoinositide 3-kinase (PI3K)-related phosphorylation of small mother against decapentaplegic (SMAD) 2/3. In turn, this would lead to LOXL1–3 activation and collagen production and cross-linking [44].

In our study, NAFLD patients had all the classic features of the metabolic syndrome, which shares an insulin resistance state, as shown by high BMI, glycemia, cholesterol, triglycerides and increased incidence of obesity and prediabetic/diabetic conditions [45]. On the contrary, AST/ALT values were slightly but significantly higher in HCV patients, which is consistent with the predominantly cytonecrotic mechanisms at the basis of HCV-related liver damages. As regards these clinical and demographic variables, our study could not demonstrate a correlation with circulating PN; in particular, no associations were found with BMI, diabetes, hypercholesterolemia and hypertriglyceridemia. Consequently, from these data, it could be stated that—at least in our casuistry—there was no pathophysiological relation between the typical NAFLD metabolic profile and soluble PN, and vice versa. These data are in contrast with those reported in the clinical Asian study conducted by Zhu et al. who showed significant positive associations between serum PN and many important metabolic parameters such as weight, fasting plasma glucose, waist circumference (WC) and hip circumference [16]. In another Chinese study (but, in this case, on overweight and obese patients), PN levels were again positively and significantly associated with WC, fasting insulin, homeostasis model assessment-insulin resistance (HOMA-IR), AST, ALT, and γ-glutamyltranspeptidase (GGT). However, similarly to our case, it was not possible to identify significant correlations with age, impaired fasting glucose, impaired glucose tolerance or BMI [17]. The authors hypothesized that it is above all the accumulation of visceral fat (which correlates well with the measurement of WC), rather than BMI, that could influence the levels of soluble PN. So, the fact that the former anthropometric parameter was not measured in our study could represent a possible limitation. It should be noted that Asian populations are notoriously subject to a greater risk of cardiovascular complications and development of type 2 diabetes mellitus—also at lower BMI values—than Caucasian populations, emphasizing the role played by racial/ethnic differences in this context, as below discussed [46,47,48]. In any case, to the best of our knowledge, there are currently no other studies besides ours which have focused on these correlations in the specific context of Caucasian populations, and that could help to better clarify the cited discrepancies.

The present study did not also identify a significant association between PN levels and the presence of NAFLD. As matter of fact, PN plasma median concentrations did not show statistically significant differences amongst the three studied groups (NAFLD, HCV, and healthy controls). Again, Chinese studies reached different conclusions, identifying a close association with NAFLD. For instance, Lu et al. found increased serum PN levels in NAFLD patients vs. normal subjects [12]. Similarly, Zhu et al. correlated increased levels of PN with the presence of NAFLD, after adjustment for age, gender, BMI and waist-to-hip ratio, even proposing a possible role of the protein as a biomarker in NAFLD management [16]. Yang et al., on the other hand, showed that, in overweight and obese patients, serum PN was associated with a higher risk for NAFLD, regardless of other risk factors such as insulin resistance [17]. Although a clear explanation for these discrepancies with our study cannot be given, it is reasonable to postulate again, that differences in populations and ethnicity, as well as in sample size, may be crucial. In this regard, it is interesting to note that—when confronted with a previous study conducted on a Caucasian population (therefore similar to ours)—comparable results were obtained with our research [19]. The authors indeed evidenced that that concentrations of circulating PN in NAFLD were not higher than in healthy controls; likewise, PN levels were not statistically different between simple steatosis subjects, NASH patients, and controls. This study is also important because it evidences how also methodological differences (e.g., the performance or not of a liver biopsy) could influence the discrepancies between our study and Asian research. The latter investigations were, in fact, conducted only on ultrasound-diagnosed NAFLDs, unlike ours and that of Polyzos et al. where the diagnosis was rigorously made with the current gold standard, i.e., liver biopsy. In this context, it therefore appears that a strength of our study includes having almost all histological diagnoses, even if the sample size was not particularly high. Moreover, it may constitute an element of innovation the fact that, to our knowledge, no comparisons existed—until now—between patients with NAFLD and other liver diseases.

Another message that emerged from our study is that in NAFLD subjects, no correlation was observed between plasma PN levels and the histological disease stage (including cirrhosis). The same applied for the HCV control group, although a trend of increased PN levels was observed in moderate fibrosis vs. no/mild fibrosis. Additionally, PN levels did not change in relation to a greater or lesser degree of steatosis (again, ascertained and quantified at liver biopsy). These particular aspects are currently not yet sufficiently clarified in the available literature. The Asian papers mentioned above simply did not explore this topic [13,16,17], while that from Polyzos et al. reported a somewhat apparently paradoxical fact: reduced PN levels, in correlation with BMI or WC, were able to predict F2 and F3 fibrosis stages with a sensitivity and specificity of 100%. With PN also being related to hepatic fibrogenesis, the authors hypothesized that down-regulation of protein expression may occur as fibrosis progresses, as a defense mechanism against further damage progression [19]. Instead, the report that PN levels were similar within other histological lesions (steatosis grade, ballooning, lobular, and portal inflammation) was in accordance with ours.

What clearly emerged from our study is, instead, that plasma levels of PN were significantly higher in patients with HCC than in all the other stages of the disease, including cirrhosis, both considering the entire study population, and NAFLD or HCV patients separately. Moreover, at multivariate analysis, HCC was an independent predictive variable for high plasma levels of PN. Our finding is in accordance with data showing that PN is not only involved in bone metabolism, but also in modulating cell proliferation and tumorigenesis in many other tissues and organs, including liver [49]. For example, Zinn et al. found that PN levels are positively correlated with tumor phenotype in glioblastoma patients and orthotopic xenografts [50]. Furthermore, higher PN expression in pancreatic neuroendocrine tumors can facilitate their revascularization by up-regulating fibroblast growth factor [51]. PN is also upregulated in non-small cell lung cancer, and its overexpression could enhance signal transducer and activator of transcription (Stat) 3 and Ak strain transforming (Akt) phosphorylation and survivin expression [52]. As regarding liver, PN, that could be released by cancer-associated fibroblasts and/or hepatic stellate cells, has been found to be associated with a high metastatic capability of HCC, being able to promote adhesion, migration and invasion of liver cancer cells, as well as angiogenesis [10]. Increased PN expression would also contribute to tumor cell survival during hypoxia through the overexpression of hypoxia-inducible factor (HIF)-1α [53]. As a likely consequence of these changes in the biological characteristics of HCCs, it was reported that tumors with PN positive expression showed more frequent multicentricity, microvascular invasion, Edmondson grade III–IV, and TNM stages III–IV than HCCs with PN negative expression. Additionally, further assessment demonstrated that overall survival and disease-free survival were better in patients without PN expression: both Kaplan–Meier and multivariate analysis showed that the expression of PN was an independent predictor of poor prognosis. For all these reasons, PN is now considered as an important prognostic biomarker of tumor recurrence and as a tool to distinguishing HCC patients from non-malignant liver diseases [10,22,54]. Overall, while taking into account the important size limits of this study with regard to the HCC arm (only 17 patients studied in total, of which 10 NAFLD), our findings seem to support the above-mentioned evidence, and would highlight the role of PN as a soluble biomarker potentially useful for the management of HCC patients, similarly to what already happens in current clinical practice for alpha-fetoprotein, as would be suggested by the good correlation between the two proteins that we found in our casuistry. It should also be noted that—although this study was not specifically designed to study HCC—it is the first to have analyzed HCCs specifically due to NAFLD. Although the data are preliminary and need further validation, it would seem that the modulation of PN circulating levels is not strictly dependent on the HCC etiology, being similar also in the HCV control group.

In consideration of the above-mentioned possible changes of PN expression in relation to ethnicity, in the present study we performed a genetic analysis by examining the main polymorphisms of *POSTN* gene in our population. In this regard it has to be said that the association between these genetic variables and chronic disease liver diseases has not been studied previously, unlike what (partially) happened for other disorders. The SNPs which have more literature solid data and, therefore, that were selected by us are rs9603226, rs3829365, and rs1028728. The most convincing research so far was conducted on Japanese patients, and investigated if these SNPs had any relationship with plasma PN concentrations and the clinical severity of bronchial asthma [23]. The study demonstrated that high PN levels were associated with worse pulmonary function tests, and that certain genotypes of rs3829365 and rs9603226 (but not of rs1028728) resulted in higher plasma PN concentrations and a more rapid decline in respiratory function, respectively. The authors hypothesized that the rs3829365 G/G genotype may be implicated in the mRNA stability of PN, consequently leading to changes in the circulating levels of the protein. The minor A allele of rs9603226 could instead up-regulate the binding capacity of the PN to other ECM proteins, thereby facilitating greater airway remodeling. Additionally, a study conducted on Chinese patients affected by heart failure showed an independent correlation between the C/G and G/G genotypes of rs3829365 and the disease, both in terms of risk and severity; rs1028728 was also not relevant in this context [55] [54]. When focusing on Caucasian subjects, however, other studies failed to confirm a predictive role of these SNPs in disease progression: for instance, rs3829365 and rs1028728, as well as their haplotype combinations, were not associated with atherosclerosis or coronary collateral circulation in subjects with coronary artery disease [24,25].

For what concerns our study, we did not find any correlation between *POSTN* polymorphisms and PN levels in NAFLD subjects; the same happened for HCV control patients. No correlations were found for the haplotypes, when analyzed separately. This is, therefore, not in agreement with what Japanese researchers found, albeit in a different disease model. However, it has to be said that our Caucasian subjects had a substantial different genetic pattern from that study population. As a matter of fact, the frequencies of the minor alleles reported by Kanemitsu et al. were 0.136, 0.278, and 0.330, for rs1028728, rs3829365, and rs9603226, respectively [23]. These frequencies were reported to be similar in Chinese subjects, taking into account the evidence provided by the aforementioned studies even if not directly focused on population genetics [13,16,17]: 0.052, 0.320, and 0.278, respectively [56,57,58]. In our population, the same frequencies were instead 0.242, 0.041, and 0.140, respectively, and comparable to what was reported in the general European population (0.263, 0.070, and 0.106, respectively) [56,57,58]. The same considerations can be made for the haplotype frequencies identified, respectively, by the Japanese researchers and us: haplotype one, 0.322 vs. 0; haplotype two, 0.278 vs. 0.596; haplotype three, 0.133 vs. 0.003; haplotype four, 0.267 vs. 0.025; haplotype five, 0 vs. 0.137; haplotype six, 0 vs. 0.239 [23].

So, ethnicity is also confirmed in our study to play an obvious major role in genetic patterns, and this could partially account for the reported by us absence of correlation with PN expression. A greater weight of genetics among Asian populations in determining protein expression and thus PN-mediated development of liver steatosis may also be hypothesized. On the contrary, in the Caucasian population, which certainly presents a greater prevalence of the major recognized risk factors for NAFLD (obesity, insulin resistance, high-fat diets, sedentary lifestyle), the role of environmental factors would be predominant in the development of the disease, and this would explain the similar results of this study and that conducted on Greek patients [19]. In addition to the previous analyses, the homo- or heterozygous diplotype was taken into consideration for the most common haplotype in the studied population (number two). New information that emerged from our study was that the subjects with this haplotype, either as homozygote or heterozygote, had lower PN levels in comparison with all other diplotypes. However, at multivariate analysis, diplotype */* (i.e., with both alleles different from haplotype two) failed to be an independent predictive variable for high plasma levels of PN in NAFLD patients. Moreover, our study evidenced, for the first time, significant differences in steatosis grading and fibrosis staging among different diplotypes, which would confirm a role of the genetic pattern of PN expression in the onset of liver disease and its severity. Hence, since up to now no information is available in the literature about this issue, our data—although needing to be validated in larger casuistries—would be of particular relevance.

This study obviously has some important limitations. First of all, only patients referring to a single center were recruited, even if it was a large tertiary referral hospital. Another issue, as said before, is related to the number of patients. Although the size of population was properly calculated by means of specific statistical power analysis, increasing the number could have improved the strength of results obtained. Moreover, only subjects with histological evidence of disease were selected for this research, which could represent both a strength and a selection bias: in any case, it should not be forgotten that in the vast majority of the previous studies NAFLD was diagnosed only by ultrasound. In addition, the cross-sectional design of the study could not prove if increased levels of protein expression at baseline were associated with an increased risk of HCC development, taking into account that it was not possible to observe a trend for PN increase in the more severe stages of NAFLD disease and that patients already with HCC were recruited at a single time point (i.e., at diagnosis). Finally, the study lacks an assessment of PN mRNA expression at the tissue level, in order to verify whether a specific expression pattern correlates with different SNP genotypes.

In conclusion, this study does not support the hypothesis that circulating PN is a useful biomarker of NAFLD in the Caucasian population (as regards both fibrosis and steatosis). However, increased plasma PN levels could be predictive for specific *POSTN* genotypes and NAFLD/NASH-derived HCC. So, PN confirms itself as a potential diagnostic and prognostic marker for HCC, also for what concerns NAFLD. To better address this issue, further studies could be organized in populations of HCC Caucasian and Asian patients stratified as concerning *POSTN* polymorphisms (haplotype), liver disease etiology and severity of the tumor disease. PN could also be quantified in the liver (from bioptic samples or resections) as protein expression and/or mRNA in order to better clarify its relations with the variables just mentioned. The comparison between tissue expression and plasma levels could also add information about the “dynamics” of PN release and its real role as a reliable circulating marker for HCC. Moreover, on the ground of the results of this study showing a correlation between plasma PN and alpha fetoprotein, it could be worth examining the correlations with other circulating HCC markers (including those being validated in liquid biopsies such as glypican-3 and glutamine synthetase) in order to strengthen the power of PN as a biomarker for this disease.

The clinical implications of the results obtained about plasma PN in HCC could be related to outcome parameters normally used by clinicians in common clinical practice (e.g., survival rate, event-free survival or reintervention rate) or to the results of different therapeutic approaches, which could involve PN itself. In this context, the role of PN could become of higher importance considering the existence of PN antagonists, aptamers, which are modified nucleic acids that specifically bind PN and inhibit its function. To date, PN antagonists have been investigated in breast and gastric cancer; however, a wider understanding of the role and mechanisms of PN in hepatic inflammation and fibrosis may make PN antagonists an innovative therapeutic approach for HCC and, maybe, also for NASH, when the stigmata of the metabolic syndrome are prevalent due to PN’s known pathophysiological role [59].

## Figures and Tables

**Figure 1 diagnostics-10-01003-f001:**
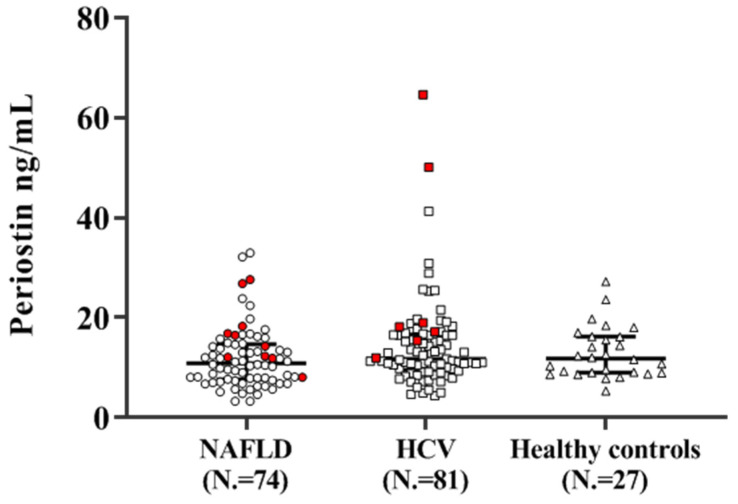
Scatterplot of the distribution of plasma periostin values according to the different disease etiologies. The red symbols identify cases of HCC. NAFLD: non-alcoholic fatty liver disease; HCV: hepatitis C virus.

**Figure 2 diagnostics-10-01003-f002:**
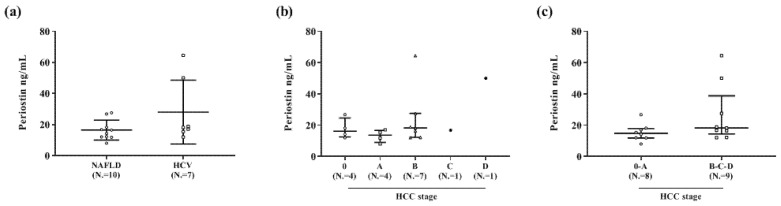
Scatterplot of the distribution of plasma periostin values in HCC patients. NAFLD: non-alcoholic fatty liver disease; HCV: hepatitis C virus. (**a**): periostin concentrations according to HCC etiology. (**b**): periostin concentrations according to HCC stages. (**c**): periostin concentrations according to combined (0–A and B–C–D) HCC stages.

**Table 1 diagnostics-10-01003-t001:** *POSTN* gene haplotypes based on the allelic variants of the three polymorphisms analyzed.

	rs9603226	rs3829365	rs1028728
Haplotype 1	A	G	A
Haplotype 2	G	C	A
Haplotype 3	G	G	T
Haplotype 4	G	G	A
Haplotype 5	A	C	A
Haplotype 6	G	C	T

*POSTN*: periostin.

**Table 2 diagnostics-10-01003-t002:** Comparison of the main demographic and clinical features of the studied population according to the etiology of liver disease. Data are presented as medians (range) for continuous variables, and as frequencies (%) for categorical variables. The *p* value refers to the comparison between NAFLD and HCV groups.

Parameter	NAFLD (N. = 74)	HCV (N. = 81)	*p*	All Patients (N. = 155)
Age, years	56 (47–66)	55 (45–64)	0.532	56 (45–66)
Male/female sex, N.	37 (50)/37 (50)	40 (49)/41 (50)	1.000	77 (50)/78 (50)
Body Mass Index, kg/m^2^	28.2 (26.8–33.2)	23.1 (21.3–27.4)	**<0.001**	27.1 (24.4–30.9)
Obese subjects, N.	31 (42)	11 (14)	**<0.001**	42 (27)
Diabetic/prediabetic subjects, N.	20 (27)/38 (51)	14 (17) 28 (35)	**0.003**	34 (22)
Liver stiffness, kPa ^1^	9.1 (7.2–13.3)	8.4 (6.5–11.6)	0.536	8.8 (6.2–12.0)
CAP ^2^, db/m	300 (265–328)	251 (214–289)	**0.032**	280 (227–311)
Advanced histological fibrosis, N. ^3^	24 (38)	24 (32)	0.593	48 (31)
Severe histological steatosis, N. ^4^	13 (20)	3 (4) ^5^	**0.006**	16 (10)
NFS, value	−0.625 (−1.534–0.468)	N.A.		N.A.
NFS > 0.676 ^6^, N.	17 (23)	N.A.		N.A.
HCC, N.	10 (14)	7 (9)	0.442	17 (11)
Platelets, ×10^9^/L	182 (153–236)	185 (144–233)	0.617	183 (146–234)
AST, U/L	36 (24–52)	56 (35–84)	**<0.001**	41 (29–67)
ALT, U/L	49 (30–75)	77 (50–113)	**<0.001**	62 (36–98)
AST/ALT ratio	0.75 (0.58–0.97)	0.70 (0.58–0.93)	0.878	0.73 (0.58–0.95)
Bilirubin, mg/dL	0.7 (0.5–1.0)	0.8 (0.6–1.1)	0.113	0.8 (0.6–1.0)
Albumin, g/L	43 (40–45)	43 (41–46)	0.904	43 (40–46)
Total cholesterol, mg/dL	170 (145–199)	158 (134–185)	**0.023**	165 (137–189)
Triglycerides, mg/dL	129 (87–170)	98 (82–121)	**0.002**	108 (83–139)
Glucose, mg/dL	114 (100–140)	100 (93–110)	**<0.001**	105 (94–117)
Creatinine, mg/dL	0.8 (0.7–0.9)	0.7 (0.7–0.9)	**0.050**	0.77 (0.67–0.90)

NAFLD: non-alcoholic fatty liver disease; HCV: hepatitis C virus; CAP: controlled attenuation parameter; NFS: NAFLD fibrosis score; HCC: hepatocellular carcinoma; AST: aspartate aminotransferase; ALT: alanine aminotransferase; N.A.: not applicable. ^1^ Available and with technically valid results in 102/155 patients (N. = 50 NAFLD and N. = 52 HCV). ^2^ Available for 25 NAFLD and 24 HCV patients. ^3^ Defined as Kleiner stage ≥ 3 or Ishak stage ≥ 4 for NAFLD and HCV subjects, respectively. HCC patients are not included. ^4^ Defined as a percentage of steatotic hepatocytes greater than 66%. HCC patients are not included. ^5^ Two HCV genotypes 1b and one genotype 3. ^6^ Predictive of severe fibrosis (Kleiner stage ≥ 3). Bold values denote statistical significance at the *p* < 0.05 level.

**Table 3 diagnostics-10-01003-t003:** Main demographic and clinical features of the healthy controls (N. = 27). Data are presented as medians (range) for continuous variables, and as frequencies (%) for categorical variables.

Parameter	Parameter
Age, years	25 (25–29)
Male/female sex, N.	11 (41)/16 (59)
Body Mass Index, kg/m^2^	21.7 (19.4–23.4)
Obese subjects, N.	0 (0)
Liver stiffness, kPa	4.4 (3.9–5.4)
CAP, db/m	196 (177–230)

CAP: controlled attenuation parameter.

**Table 4 diagnostics-10-01003-t004:** (a): plasma periostin concentrations according to the severity of liver disease. Data are presented as medians (range). The *p* value refers to the differences between the four considered disease stages. (b): differences of PN circulating levels within the various fibrosis stages in non-HCC individuals. *p* values are reported. (c): differences of PN circulating levels between HCC and non-HCC patients divided according to the various fibrosis stages. *p* values are reported.

**(a)**		**Plasma Periostin (ng/mL)**				
		**No/mild fibrosis (N. = 79) ^1^**	**Moderate fibrosis (N. = 38) ^2^**	**Severe fibrosis (N. = 48) ^3^**	**HCC** **(N. = 17)**	***p***
	Total population (N. = 182)	10.5 (8.1–15.0)	12.0 (10.8–15.5)	11.1 (8.4–15.7)	16.8 (12.2–22.9)	**0.001**
	NAFLD (N. = 74)	9.9 (6.8–13.1)	10.9 (7.1–11.3)	10.2 (7.7–14.7)	15.4 (12.1–18.3)	**0.045**
	HCV (N. = 81)	10.4 (7.8–14.4)	13.1 (10.9–16.5)	11.4 (8.9–18.2)	18.2 (15.5–50.1)	**0.013**
	Healthy controls (N. = 27)	12.0 (9.1–16.3)	-	-	-	N.A.
**(b)**		**Differences in plasma periostin concentrations, *p* value**				
		**Moderate vs. no/mild fibrosis ^1,2^**	**Severe vs. no/mild fibrosis ^1,3^**	**Severe vs. moderate fibrosis ^2,3^**	**Cirrhosis ^4^ vs. pre-cirrhosis ^5^**	
	Total non-HCC subjects (N. = 165)	0.101	0.424	0.548	0.937	
	Non-HCC NAFLD (N. = 64)	0.952	0.535	0.696	0.238	
	Non-HCC HCV (N. = 74)	0.073	0.254	0.889	0.535	
**(c)**		**Differences in plasma periostin concentrations, *p* value**				
		**HCC vs. no/mild fibrosis ^1^**	**HCC vs. moderate fibrosis ^2^**	**HCC vs. severe fibrosis ^3^**		
	Total patients (N. = 155)	**<0.001**	**0.002**	**0.004**		
	NAFLD (N. = 74)	**0.008**	**0.01**	**0.04**		
	HCV(N. = 81)	**0.007**	**0.004**	**0.03**		

NAFLD: non-alcoholic fatty liver disease; HCV: hepatitis C virus; HCC: hepatocellular carcinoma; N.A.: not applicable. ^1^ Defined as Kleiner stage ≤ 1 or Ishak stage ≤ 1 for NAFLD and HCV subjects, respectively. Number of patients: 34 NAFLD, 18 HCV. ^2^ Defined as Kleiner stage = 2 or Ishak stages 2–3 for NAFLD and HCV subjects, respectively. Number of patients: 6 NAFLD, 32 HCV. ^3^ Defined as Kleiner stage ≥ 3 or Ishak stage ≥ 4 for NAFLD and HCV subjects, respectively. Number of patients: 24 NAFLD, 24 HCV. ^4^ Defined as Kleiner stage 4 or Ishak stage 6 for NAFLD and HCV subjects, respectively. Number of patients: 12 NAFLD, 4 HCV. ^5^ Defined as Kleiner stage 3 or Ishak stages 4–5 for NAFLD and HCV subjects, respectively. Number of patients: 12 NAFLD, 20 HCV. Bold values denote statistical significance at the *p* < 0.05 level.

**Table 5 diagnostics-10-01003-t005:** Distribution of *POSTN* gene polymorphisms in the total studied population and in the three distinct groups. Data are presented as frequencies (%). The *p* value refers to the comparison between genotypes in the different study groups.

SNPs	Genotype			*p*
**rs9603226—chr13:37569449 (GRCh38.p12)**	**G/G**	**G/A**	**A/A**	
Total (N. = 182) NAFLD (N. = 74) HCV (N. = 81) Healthy controls (N. = 27)	132 (72) 53 (72) 58 (72 )21 (78)	49 (27) 21 (28) 23 (28) 5 (18)	1 (1) 0 (0) 0 (0) 1 (4)	0.153
**rs3829365—chr13:37598759 (GRCh38.p12)**	**C/C**	**C/G**	**G/G**	
Total (N. = 182) NAFLD (N. = 74) HCV (N. = 81) Healthy controls (N. = 27)	168 (92) 70 (95) 71 (88) 27 (100)	13 (7) 4 (5) 9 (11) 0 (0)	1 (1) 0 (0) 1 (1) 0 (0)	0.224
**rs1028728—chr13:37599679 (GRCh38.p12)**	**A/A**	**A/T**	**T/T**	
Total (N. = 182) NAFLD (N. = 74) HCV (N. = 81) Healthy controls (N. = 27)	106 (58) 44 (59) 46 (57) 16 (59)	64 (35) 25 (34) 29 (36) 10 (37)	12 (7) 5 (7) 6 (7) 1 (4)	0.968

POSTN: periostin; SNP: single nucleotide polymorphism; NAFLD: non-alcoholic fatty liver disease; HCV: hepatitis C virus.

**Table 6 diagnostics-10-01003-t006:** Periostin plasmatic concentrations according to *POSTN* gene polymorphisms. Median and interquartile range are reported. The *p* value refers to the correlation between the genotype of the single polymorphisms and periostin circulating levels.

SNPs	Genotype and Plasmatic Periostin (ng/mL)			
**rs9603226**	**G/G**	**G/A**	**A/A**	***p***
Total (N. = 182) NAFLD (N. = 74) HCV (N. = 81) Healthy controls (N. = 27)	11.9 (9.0–16.3) 11.4 (7.7–15.8) 12.2 (9.8–16.5) 12.0 (9.3–16.2)	11.1 (8.2–15.6) 9.7 (8.0–12.3) 11.2 (9.2–16.8) 15.6 (8.7–19.8)	9.1 (9.1–9.1) – – 9.1 (9.1–9.1)	0.491 0.232 0.742 0.710
**rs3829365**	**C/C**	**C/G**	**G/G**	***p***
Total (N. = 182) NAFLD (N. = 74) HCV (N. = 81) Healthy controls (N. = 27)	11.8 (8.6–16.4) 11.2 (7.7–14.8) 12.5 (10.1–16.8) 12.0 (9.1–16.3)	10.5 (8.7–13.0) 9.6 (7.2–11.7) 10.5 (9.2–13.2) –	64.6 (64.6–64.6) – 64.6 (64.6–64.6) –	0.121 0.430 0.119 NA
**rs1028728**	**A/A**	**A/T**	**T/T**	***p***
Total (N. = 182) NAFLD (N. = 74) HCV (N. = 81) Healthy controls (N. = 27)	11.4 (8.0–15.6) 9.8 (6.9–13.3) 12.3 (9.2–16.5) 13.3 (10.2–16.2)	11.1 (9.11–15.9) 11.9 (9.4–15.8) 11.2 (10.3–16.5) 9.2 (8.7–12.6)	15.1 (10.3–19.1) 11.8 (8.1–17.7) 16.0 (13.1–19.4) 17.1 (17.1–17.1)	0.153 0.206 0.394 0.232

POSTN: periostin; SNP: single nucleotide polymorphism; NAFLD: non-alcoholic fatty liver disease; HCV: hepatitis C virus.

**Table 7 diagnostics-10-01003-t007:** Periostin levels as a function of *POSTN* gene SNPs in NAFLD and HCV patients (N. = 155), see text for details. Median and interquartile range are reported. The *p* value refers to the differences of periostin plasma levels between the various *POSTN* genotypes. (a): dominant model (MM versus Mm + mm). (b): recessive model (MM + Mm versus mm). * = any of the two possible nucleotides for each of the conditions of interest.

	*POSTN* SNPs	*POSTN* Genotypes ^1^ and Periostin Levels ng/mL		*p*
**(a)**	**rs9603226**	**G/G**	**A/***	0.301
		11.9 (9.0–16.3)	11.1 (8.2–16.6)	
	**rs3829365**	**C/C**	**G/***	0.521
		11.8 (8.6–16.4)	10.5 (8.7–13.1)	
	**rs1028728**	**A/A**	**T/***	0.153
		11.5 (8.0–15.6)	11.9 (9.1–16.7)	
**(b)**	**rs9603226**	**G/***	**A/A**	0.714
		11.7 (9.7–16.3)	9.1 (9.1–9.1)	
	**rs3829365**	**C/***	**G/G**	0.085
		11.5 (8.7–16.2)	64.6 (64.6–64.6)	
	**rs1028728**	**A/***	**T/T**	0.091
		11.3 (8.6–15.8)	15.1 (10.3–19.1)	

POSTN: periostin; SNP: single nucleotide polymorphism. ^1^ Dominant and recessive models were calculated by referring to the allele frequency, considering the most frequent allele as the major one and the least frequent allele as the minor one.

**Table 8 diagnostics-10-01003-t008:** Plasma periostin values as a function of the *POSTN* gene diplotypes. * = any other haplotype different from haplotype 2. Median and interquartile range are reported. The *p* value refers to the differences of periostin plasma levels between the various *POSTN* diplotypes.

	*POSTN* Gene Diplotypes and Periostin Levels ng/mL			*p*
**Group**	**Diplotype 2/2**	**Diplotype 2/***	**Diplotype */***	
**Total (N. = 182)**	11.7 (7.9–16.2) (N. = 61)	11.1 (8.7–15.6) (N. = 95)	11.9 (8.9–18.8) (N. = 26)	0.253
**NAFLD (N. = 74)**	10.4 (6.8–15.0) (N. = 25)	11.1 (8.1–14.2) (N. = 39)	11.2 (8.1–17.7) (N. = 10)	0.447
**HCV (N. = 81)**	11.7 (9.3–16.5) (N. = 25)	11.2 (9.2–16.5) (N. = 43)	13.2 (11.0–19.4) (N. = 13)	0.233
**Healthy controls (N. = 27)**	12.4 (10.8–16.2) (N. = 11)	10.4 (9.0–18.1) (N. = 13)	9.1 (8.7–17.1) (N. = 3)	0.460

POSTN: periostin; NAFLD: non-alcoholic fatty liver disease; HCV: hepatitis C virus.

**Table 9 diagnostics-10-01003-t009:** Predictive parameters of high plasma periostin (i.e., last quintile) values. Both odds ratios (OR) with the corresponding 95% confidence intervals (CI) and statistical significance (*p* values) are reported of a maximum-likelihood logit regression model with high (i.e., belonging to the last quintile) plasma periostin as dependent variable, and *POSTN* diplotype (categorized as 2/2 plus 2/* versus */*), age, sex, HCC, and platelet count as predictor variables. (a): all patients. (b): NAFLD patients. (c): HCV patients.

	(a) All Patients (N. = 155)		(b) NAFLD (N. = 74)		(c) HCV (N. = 81)	
Variable	OR (95% CI)	*p*	OR (95% CI)	*p*	OR (95% CI)	*p*
**Diplotype 2/2 + 2/***	0.25 (0.09–0.70)	**0.008**	0.37 (0.07–2.00)	0.219	0.25 (0.07–0.97)	**0.041**
**Age, years**	0.98 (0.95–1.02)	0.338	0.97 (0.92–1.03)	0.342	0.99 (0.95–1.04)	0.808
**Male/female sex**	1.98 (0.79–4.96)	0.144	1.67 (0.39–7.11)	0.519	2.74 (0.80–9.35)	0.113
**HCC, N.**	6.76 (1.73–26.38)	**0.006**	7.46 (1.19–56.02)	**0.022**	15.64 (2.15–113.87)	**0.001**
**Platelets, ×10^9^/L**	0.99 (0.98–1.00)	0.230	1.00 (0.99–1.01)	0.399	1.00 (0.99–1.01)	0.883

NAFLD: non-alcoholic fatty liver disease; HCV: hepatitis C virus; OR: odds ratio; CI: confidence interval; HCC: hepatocellular carcinoma. Bold values denote statistical significance at the *p* < 0.05 level.

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
