# Peer review of "Periostin Circulating Levels and Genetic Variants in Patients with Non-Alcoholic Fatty Liver Disease"

_diagnostics, 2020, doi:10.3390/diagnostics10121003_

Round 1
Reviewer 1 Report
In this interesting manuscript, Smirne, et al., have described results from an observational study where they have investigated periostin levels and genetic variations in NAFLD, HCV and HCC patients. They have shown that high levels of periostin is not correlated with NAFLD prevelance (including cirrhosis). Furthermore, they have shown that hepatocellular carcinoma is the sole predictor of high circulating concentrations of periostin.
Some comments -
1. please stratify all patient results by sex. With 155 patients with at least 37 patients in each sex, it will be very interesting to see how the periostin levels compare when stratified by sex.
2. lines 181 – 185: it is unclear where the data referred to by the text is presented. If this is not shown in the manuscript, please include a table/figure for the data.
3. lines 273 – 275: This very interesting observation is not discussed further. Perhaps, additional discussion could be added in the discussion section juxtaposed with what is available in the literature.
4. lines 301 – 313: please move this to the introduction. This text describing periostin in detail is better suited in the introduction.
5. What multivariate analysis was carried out in this study? The section on “Statistical analysis” as written currently does not make it clear as to which test was performed on each set data presented. Please include details on statistical analyses in the figure caption and expand the section under methods to include more details.
6. Was any co-variate used for statistical analysis (e.g. ANCOVA) when comparing periostin levels across NAFLD, HCV, and healthy subjects? For example, ANCOVA analysis with triglycerides (TG) as a covariate might be interesting since NAFLD development is often characterized by TG accumulation.
7. Please refer to earlier works with author names (for example, Jones, et al., or Lu, et al.) instead of the current colloquial references.
Author Response
Manuscript ID: diagnostics-968192
Title: Periostin circulating levels and genetic variants in patients with non-alcoholic fatty liver disease
Subject: Point-by-point response to the reviewer 1 comments
November 16, 2020
Dear Sirs,
hereby please find our point-by-point response to the reviewer 1 comments. Thank you very much for your attention.
Sincerely,
Carlo Smirne, MD, PhD
Important note to reviewer and editor: The Journal required authors to download the latest version of the manuscript for revision from Journal website, because the original submission may have been changed. Since this version completely lost the formatting compared to the one originally submitted (e.g. as regards all characters in italics or in superscript, and all hypertext links to the reference editing program used for manuscript editing), we decided, for your clarity, not to use the "Track Changes" function in Microsoft Word (which would have shown many changes -compared to the initial manuscript- which actually had not been made), but only to highlight in yellow all the real changes with respect to the initial submitted version, as detailed in the point-by-point discussion below. The line numbers and reference citations always refer to the manuscript as initially submitted and downloadable from the journal website as described above.
In this interesting manuscript, Smirne, et al., have described results from an observational study where they have investigated periostin levels and genetic variations in NAFLD, HCV and HCC patients. They have shown that high levels of periostin is not correlated with NAFLD prevalence (including cirrhosis). Furthermore, they have shown that hepatocellular carcinoma is the sole predictor of high circulating concentrations of periostin.
Some comments:
1. please stratify all patient results by sex. With 155 patients with at least 37 patients in each sex, it will be very interesting to see how the periostin levels compare when stratified by sex.
Considering all patients (N.=155), differences among sexes with regard to PN levels did not reach statistical significance, either considering them as a whole (median in males: 11.9 ng/mL, interquartile range 8.2-16.8; median in females 11.1 ng/mL, interquartile range 8.5-14.8, p = 0.196), or analyzing separately two subgroups, i.e. HCV (median in males: 12.9 ng/mL, interquartile range 10.7-17.4; median in females 11.3 ng/mL, interquartile range 9.2-16.0, p = 0.275) and NAFLD (median in males: 11.8 ng/mL, interquartile range 7.8-16.1; median in females 10.5 ng/mL, interquartile range 7.1-13.3, p = 0.418). This new piece of
information has been added in the Results at the 2.2.1 section (Periostin concentrations and disease etiology) after line 151 of the original manuscript.
2. lines 181 – 185: it is unclear where the data referred to by the text is presented. If this is not shown in the manuscript, please include a table/figure for the data.
A new Figure (Figure 2) has now been added showing the scatterplot of the distribution of plasma periostin values according to the two different etiologies of HCC, and amongst different HCC stages (either separately or combined as specified in the text). The correct references to the figure have been added in the text (lines 181-185 of the original manuscript). Hereby the new Figure is shown:
Figure 2. Scatterplot of the distribution of plasma periostin values in HCC patients. NAFLD: non-alcoholic fatty liver disease; HCV: hepatitis C virus. (a): periostin concentrations according to HCC etiology. (b): periostin concentrations according to HCC stages. (c): periostin concentrations according to combined (0-A and B-C-D) HCC stages.
3. lines 273 – 275: This very interesting observation is not discussed further. Perhaps, additional discussion could be added in the discussion section juxtaposed with what is available in the literature.
No information is available in the literature for what concerns periostin diplotypes, neither with regard to liver diseases nor, more in general, for any other model of disease. As a matter of fact, to best of our knowledge, periostin genetics so far has been studied only for what concerns haplotypes. The researches which have been published and that can be found in the Publications section of Single Nucleotide Polymorphism Database (dbSNP) of NIH are:
- 1. BMC Cardiovasc Disord. 2015 May 12;15:37. doi: 10.1186/s12872-015-0027-z.
- 2. Arterioscler Thromb Vasc Biol. 2011 Jul;31(7):1661-7.
- 3. J Cardiovasc Med (Hagerstown). 2011 Jul;12(7):469-74. doi: 10.2459/JCM.0b013e328347e48c.
- 4. Allergol Int. 2014 Jun;63(2):181-8. doi: 10.2332/allergolint.13-RA-0670.
- 5. Osteoporos Int. 2012 Jul;23(7):1877-87. doi: 10.1007/s00198-011-1861-1. Epub 2012 Jan 4.
Paper number 4 expands data previously reported by the same group in J. Allergy Clin. Immunol. 2013, 132, 305–312, the latter one being extensively discussed in our paper. However, none of these papers specifically deals with liver diseases: evidence is limited mainly to the cardiology and pulmonology fields, as stated in the Introduction section. In other words, the correlation between periostin genetics and liver diseases is unprecedented in the literature. We decided in any case to expand the Discussion section stressing the importance of the innovative results we obtained on the correlation of periostin diplotypes and histological damage in the liver. The following statements have now been added from line 469 of the original manuscript:
Moreover, our study evidenced, for the first time, significant differences in steatosis grading and fibrosis staging among different diplotypes, which would confirm a somehow role of the genetic pattern of periostin expression in the onset of liver disease and its severity. Hence, since up to now no information is available in the literature about this issue, our data -although needing to be validated in larger casuistries- would be of particular relevance.
4. lines 301 – 313: please move this to the introduction. This text describing periostin in detail is better suited in the introduction.
Lines 301-313 have been moved to the introduction starting from line 45 of the original manuscript. The new Introduction paragraphs (reference numbers are with the original numbering for your convenience, but have been renumbered in the revised manuscript; a new reference has been added and the complete citation is hereby shown, replaced in the revised text by the correct numbering) are presented below, including a new paragraph on how PN down-regulates PPAR alpha expression and thus fatty acid beta-oxidation, as requested by reviewer #2:
PN, also known as osteoblast-specific factor 2 (OSF-2), is a 90 kDa multifunctional extracellular matrix (ECM) protein, coded by periostin (POSTN) gene and mainly secreted by osteoblasts [6]. It has pleiotropic activities far beyond simple bone remodeling; as a matter of fact it is also involved, amongst others, in the pathophysiology of arthritis, atherosclerosis, and inflammatory diseases [24]. Actually, one of the target organs in which PN has been shown to play a crucial role is the liver, where it can modulate the cell fate determination and proliferation, inflammatory responses, ECM remodeling, even tumorigenesis [7–11]. In this respect, the strongest evidence so far concerns its pivotal role in the onset of metabolic disease (such as obesity and glucose or lipid disorders) by suppression of fatty acid oxidation in the liver [12]. As a matter of fact, in obese mice the overexpression of PN in the liver was shown to induce hepatic steatosis and hypertriglyceridemia through the downregulation of peroxisome proliferator-activated receptor (PPAR)-α, which activates the fatty acid oxidation in mitochondria and peroxisomes [25]. Conversely, the genetic knockout of PN significantly improved those conditions [12] and was able to protect mice against dietary-induced NAFLD [26]. In detail, PN could exert its protective effects against hypertriglyceridemia and liver steatosis through the interaction with the subtype α6β4 of the integrins family and the subsequent activation of an intracellular pathway involving Ras-related C3 botulinum toxin substrate (Rac) 1 and c-JUN. Those events would lead to the inhibition of the PPAR-α promoter, RAR-related orphan receptor (ROR) α, and to the downregulation of PPAR-α itself [12].
Considering human studies, the relationship between PN and hepatic steatosis, which is the basis of the present study, was first proposed by Lu, et al. [12], showing an upregulation of hepatic PN expression in NAFLD patients, with a good correlation with hepatic TG content. Moreover, serum PN levels were found increased in the same subjects, although without a significant correlation with hepatic TGs. Also in the studies from Zhu, et al. and Yang, et al. higher plasma PN concentrations were observed in NAFLD patients in comparison to healthy controls, although without significant differences between the ultrasound degrees of NAFLD [13,14]. In addition, circulating PN was strongly associated with lipid metabolism, chronic inflammation and insulin resistance [14,27], and the serum and liver tissue levels of PN were closely related to the decline of liver function and to the pathological stage in NAFLD patients [7,14]. It is to note that plasma PN analysis performed through the same Enzyme Linked Immunosorbent Assay (ELISA) kit of the research from Zhu, et al. on Chinese subjects failed to find out any association between circulating PN and hepatic steatosis on a small but histologically-confirmed casuistry of NAFLD European patients, while there was a statistically significant trend for PN level decrease in more severe fibrosis stages [15].
5. What multivariate analysis was carried out in this study? The section on “Statistical analysis” as written currently does not make it clear as to which test was performed on each set data presented. Please include details on statistical analyses in the figure caption and expand the section under methods to include more details.
To identify independent predictors of a high periostin value (defined as a value in the last quintile for the study population), we built a maximum-likelihood logit regression model, with high periostin as dependent variable and POSTN diplotype (categorized as 2/2 plus 2/* versus */*), age, sex, HCC, and platelet count as predictor variables.
Statistical analysis section has been detailed as requested starting from line 606 of the original manuscript [“Finally, a set of variables (POSTN diplotype, age, sex, HCC, and platelet count) was analyzed in a maximum-likelihood logit regression model as a predictor of high plasma PN values (defined by belonging to the last quintile of the study population)”], and the same applies to Table 8 figure caption starting from line 288 of the original manuscript (“…Both odds ratios (OR) with the corresponding 95% confidence intervals (CI) and statistical significance (p values) are reported of a maximum-likelihood logit regression model with high (i.e. belonging to the last quintile) plasma periostin as dependent variable, and POSTN diplotype (categorized as 2/2 plus 2/* versus */*), age, sex, HCC, and platelet count as predictor variables.”).
6. Was any co-variate used for statistical analysis (e.g. ANCOVA) when comparing periostin levels across NAFLD, HCV, and healthy subjects? For example, ANCOVA analysis with triglycerides (TG) as a covariate might be interesting since NAFLD development is often characterized by TG accumulation.
When adjusted for the concentrations of serum glucose and serum triglycerides, serum periostin levels (after logarithmic transformation) where significantly higher in the HCV group than in the NAFLD group (geometric mean±SD: 12.9±1.6 vs. 10.2±2.5, respectively; p <0.001).
Statistical analysis section has been detailed starting from line 605 of the original manuscript [“The differences in a dependent variable (e.g. plasma PN values) among groups, taking into account its variability explained by one or more covariates, were tested through the analysis of covariance (ANCOVA)”], and the same applies starting from line 218 of the original manuscript [“Finally, when adjusted for the concentrations of serum glucose and serum TGs, plasma PN levels (after logarithmic transformation) where significantly higher in the HCV group (N.=74) than in the NAFLD group (N.=64) (geometric mean±standard deviation (SD): 12.9±1.6 vs. 10.2±2.5 ng/mL, respectively; p <0.001)”].
7. Please refer to earlier works with author names (for example, Jones, et al., or Lu, et al.) instead of the current colloquial references.
Citations of earlier works with author names have now all been modified as suggested.
8. Extensive editing of English language and style required.
English in the manuscript was thoroughly rechecked and edited for language and form.

Reviewer 2 Report
This exciting study reveals the limitations of periostin as a serum marker of liver disease and a possible therapeutic target for treatment. This is a well-performed study.
You need to include a paragraph on the believed mechanism of periostin's action and possibly a figure showing the function in the liver matrisome.
Why weren't other serum markers of the liver disease reported in this study, such as glypican-3, glutamine synthetase, alpha-fetoprotein, etc., which would both supportive and comparative data for your assertion that periostin is a possible marker of advance HCC? In this context, have you observed increased periostin in more advanced HCC (Barcelona stage D vs. stage A)?
Line 78 spell out the genotypes for the four major and two minor haplotypes.
Table 3 includes HCC NAFLD disease periostin serum levels to compare with increased levels observed in HCV HCC, or is HCC shown reporting both HCV and NAFLD HCC?
Table 4, can you add to each reference polymorphism (rs) the POSTN gene's location? Also, can you identify and clarify the rs associated with each haplotype and diplotype? This isn't easy to follow.
It would be appropriate and informative if you include the diplotype of the control population for comparative purposes in tables 7 and 8.
Can you provide further information on how PN down-regulates PPAR alpha expression and thus fatty acid beta-oxidation? Line 308.
Line 355, you seem to have explained the difference observed in your study with Caucasian population and previous studies with Asian people that showed serum PN levels correlated with liver disease progression. Can you expand this discussion to include the significant differences in the genotypes of the Asian and Caucasian population used in these studies? Line 456-469.
Line 395 to 421 excellent discussion. You should provide a summary paragraph on where PN studies should be directed further concerning its use as a marker of HCC or advanced liver disease in different ethnic populations.
Author Response
Manuscript ID: diagnostics-968192
Title: Periostin circulating levels and genetic variants in patients with non-alcoholic fatty liver disease
Subject: Point-by-point response to the reviewer 2 comments
November 16, 2020
Dear Sirs,
hereby please find our point-by-point response to the reviewer 2 comments. Thank you very much for your attention.
Sincerely,
Carlo Smirne, MD, PhD
Important note to reviewer and editor: The Journal required authors to download the latest version of the manuscript for revision from Journal website, because the original submission may have been changed. Since this version completely lost the formatting compared to the one originally submitted (e.g. as regards all characters in italics or in superscript, and all hypertext links to the reference editing program used for manuscript editing), we decided, for your clarity, not to use the "Track Changes" function in Microsoft Word (which would have shown many changes -compared to the initial manuscript- which actually had not been made), but only to highlight in yellow all the real changes with respect to the initial submitted version, as detailed in the point-by-point discussion below. For what concerns the following discussion, our comments are in italics, the extracts of the manuscript text in block letters. The line numbers always refer to the manuscript as initially submitted and downloadable from the journal website as described above; a new reference citation has now been added (please see point 1b), and is presented as full reference in this discussion.
This exciting study reveals the limitations of periostin as a serum marker of liver disease and a possible therapeutic target for treatment. This is a well-performed study.
You need to include a paragraph on the believed mechanism of periostin's action and possibly a figure showing the function in the liver matrisome.
We have added more information and new references about the possible mechanisms of action of periostin, which could involve liver matrisome, as well.
a) From line 309 of the original manuscript (but actually moved together with the whole paragraph in the introduction section, as requested by reviewer 1: “…This text describing periostin in detail is better suited in the introduction“):
In detail, PN could exert its protective effects against hypertriglyceridemia and liver steatosis through the interaction with the subtype α6β4 of the integrins family and the subsequent activation of an intracellular
pathway involving Rac1 and c-JUN. Those events would lead to the inhibition of the PPAR-α promoter, ROR α, and the downregulation of PPAR-α [12].
b) From line 328 of the original manuscript:
The role of PN as a profibrotic agent through the liver matrisome modulation could be related to its direct actions on key molecules or proteins of ECM or on enzymes involved in the collagen stabilization and matrix integrity and elasticity, such as Lysyl oxidases (LOX) and LOX like enzymes (LOXL). Hence, PN was shown to cause collagen expression in HSCs via integrin αvβ3 and the activation of the Phosphoinositide 3-kinase (PI3K)-related phosphorylation of Small mother against decapentaplegic (SMAD) 2/3 which, in turn, would lead to LOXL1–3 activation and collagen production and cross-linking. [Kumar P, Smith T, Raeman R, Chopyk DM, Brink H, Liu Y, Sulchek T, Anania FA. Periostin promotes liver fibrogenesis by activating lysyl oxidase in hepatic stellate cells. J Biol Chem. 2018 Aug 17;293(33):12781-12792].- 2. Why weren't other serum markers of the liver disease reported in this study, such as glypican-3, glutamine synthetase, alpha-fetoprotein, etc., which would both supportive and comparative data for your assertion that periostin is a possible marker of advanced HCC? In this context, have you observed increased periostin in more advanced HCC (Barcelona stage D vs. stage A)?
Both circulating glypican-3 or glutamine synthetase are not routinely dosed in HCC patients diagnosed in our Hospital, and they were not part of the original research project of this study. However, we cited both proteins in a new Discussion paragraph as promising new markers for liquid biopsies (please see point 9 of this point-by-point discussion).
For what concerns alpha-fetoprotein, it was dosed, as clinical practice, in all HCC patients at the same time of periostin sampling. The concentrations (median, interquartile range) were as follows in the different BCLC stages: 31.9 (8.9-79.8), 18.7 (2.6-672.0), 19.9 (4.0-601.0), 276.3 (276.3-276.3), 9.2 (9.2-9.2) ng/mL, for stages 0, A, B, C and D, respectively. No significant differences were found within total HCC population (p= 0.834) or when HCC stages were grouped as 0-A [23.1 (4.5-79.8) ng/mL] and B-C-D [19.9 (4.7-438.6) ng/mL] (p=0.704), as performed for the analysis of PN. Finally, a correlation was observed between circulating periostin and alpha-fetoprotein levels (Spearman Rho=- 0.487, p=0.047). The following paragraph has been added starting from line 184 of the original manuscript:
Always with regard to HCC patients, a further analysis was conducted on serum alpha-fetoprotein, which was dosed, as clinical practice, in all HCC patients at the same time of PN sampling. The concentrations were as follows in the different BCLC stages: 31.9 (8.9-79.8), 18.7 (2.6-672.0), 19.9 (4.0-601.0), 276.3 (276.3-276.3), and 9.2 (9.2-9.2) ng/mL, for stages 0, A, B, C and D, respectively. No significant differences were found in alpha-fetoprotein concentrations within total HCC population (p= 0.834) or when HCC stages were grouped as 0-A [23.1 (4.5-79.8) ng/mL] and B-C-D [19.9 (4.7-438.6) ng/mL] (p=0.704), as performed for the analysis of PN. Finally, a correlation was observed between circulating PN and alpha-fetoprotein levels (Spearman Rho=- 0.487, p=0.047).
Conclusions starting from line 418 of the original manuscript have also been modified as follows:
Overall, while taking into account the important size limits of this study with regard to the HCC arm (only 17 patients studied in total, of which 10 NAFLD), our findings seem to support the above-mentioned evidences, and would highlight the role of PN as a soluble biomarker potentially useful for the management of HCC patients, similarly to what already happens in current clinical practice for alpha-fetoprotein, as it would be suggested by the good correlation between the two proteins that we found in our casuistry.
For what concerns possible increased periostin levels in more advanced HCC, original manuscript already includes this piece of information at lines 182-185:
Also, no differences were found amongst various HCC stages (p=0.09), although there was a trend for increased PN levels in more advanced stages: 14.9 (11.9-17.6) vs. 18.3 (14.4-38.8) ng/mL in stages 0-A vs. B-C-D, respectively”. In particular, with all caution due to the small numbers, periostin concentrations were not significantly different between BCLC stage D [50.1 (50.1-50.1) ng/mL] and A [13.7 (10.0-16.3) ng/mL] cases, p=1 at Mann-Whitney test. The same analysis was conducted between HCC stage D and A in the HCV subpopulation (being the D stage in our casuistry represented by a single HCV patient): again no significant differences could be found [50.1 (50.1-50.1) vs. 16.3 (16.3-16.3) ng/mL, respectively], p=1 at Mann-Whitney test.
3. Line 78 spell out the genotypes for the four major and two minor haplotypes.
The original paper from Kanemitsu et al. (J. Allergy Clin. Immunol. 2013, 132, 305–312) states as follows: “A total of 47 single-nucleotide polymorphisms (SNPs) in the region of the POSTN gene and its upstream, total 39 kb, was captured in the HapMap Japanese data set. Haplotype analysis identified 4 major haplotypes and 2 minor haplotypes. Two minor haplotypes were grouped into the closest major haplotype, and 3 tag SNPs that determined the 4 haplotypes were identified”. The 3 tag SNPs were rs1028728, rs3829365, and rs9603226. POSTN gene haplotypes based on the allelic variants of the three polymorphisms analyzed were (Figure 1 of that article):
No mention of the two minor haplotypes is made in the article. including Supplemental Materials, and the data is no more collectable from the HapMap Resource site because it has been closed by NCBI on June 16, 2016 (the official explanation is that “a recent computer security audit has revealed security flaws in the legacy HapMap site that require NCBI to take it down immediately…” as reported at https://www.ncbi.nlm.nih.gov/variation/news/NCBI_retiring_HapMap/
For all these considerations, we spelled out only the four major haplotypes (since both the minor ones were already grouped with the major ones by the Japanese researchers, as specified above, so they are not clinically relevant in this context). These haplotypes, as shown in Table 9, coincide with 4 of the 6 that we found in our research, though, obviously, at very different frequencies between the two studied populations. Starting from line 78 of the original manuscript, text has been changed as follows:
In particular, 4 major and 2 minor haplotypes could be identified (initially, in the Japanese population). The former ones were mainly determined by three tag single nucleotide polymorphisms (SNPs), which were located at intron 66 bp upstream of exon 21 (rs9603226, alleles G>A), 5’ untranslated region (rs3829365, alleles G>C), and at promoter region (rs1028728, alleles A>T) [19]. The four major POSTN haplotypes obtained respectively from these three SNPs were: AGA (haplotype 1), GCA (haplotype 2), GGT (haplotype 3), and GGA (haplotype 4).
4. Table 3 includes HCC NAFLD disease periostin serum levels to compare with increased levels observed in HCV HCC, or is HCC shown reporting both HCV and NAFLD HCC?
Periostin plasma levels in HCC patients are described in Table 3 panels a and c (panel b refers only to non-HCC subjects). In both panels a and c HCC patients are first shown in aggregate form (i.e. NAFLD plus HCV: second line of each panel) and then by individual etiology (i.e. NAFLD: third line of each panel, and HCV: fourth line of each panel). However, p values do not refer to comparisons between periostin levels in HCC patients (in aggregate or individual forms); for what concerns panel a they refer to the differences between the four considered disease stages (i.e. no/mild fibrosis; moderate fibrosis; severe fibrosis; HCC) and for what concerns panel c they refer to the differences between HCC and each of the three considered fibrosis stages (i.e. no/mild fibrosis; moderate fibrosis; severe fibrosis). Instead, the differences between periostin levels in HCC NAFLD vs. HCV patients are shown in Figure 1 (red dots) and then analyzed in lines 181-182: “As regards HCC patients, no significant differences were found between NAFLD (N.=10) and HCV (N.=7) subjects (p=0.187)”. Please note that, in order to respond to reviewer 1, a new Figure 2 has been added, consider only panel a for what concerns this item.
Figure 2. Scatterplot of the distribution of plasma periostin values in HCC patients. NAFLD: non-alcoholic fatty liver disease; HCV: hepatitis C virus. (a): periostin concentrations according to HCC etiology. (b): periostin concentrations according to HCC stages. (c): periostin concentrations according to combined (0-A and B-C-D) HCC stages.
5. Table 4, can you add to each reference polymorphism (rs) the POSTN gene's location? Also, can you identify and clarify the rs associated with each haplotype and diplotype? This isn't easy to follow.
a) In Table 4 the specific locus of each SNP was added, and an appropriate reference was made in the text. In particular, the loci were for rs9603226: chr13:37569449 (GRCh38.p12), for rs3829365: chr13:37598759 (GRCh38.p12), and for rs1028728: chr13:37599679 (GRCh38.p12). The loci were obtained from the website: www.ncbi.nlm.nih.gov/snp
b) The haplotype full sequences are shown in Table 9 belonging to Materials and Methods section, a citation has been added in the text to make it easier to find the necessary piece of information.
c) For what concerns the rs associated with each diplotype, we added a new Table in Supplementary Materials. A citation has been added in the text.
Table 2. POSTN gene diplotypes based on the 6 haplotype combinations found in the studied population. One out of the two haplotypes present on the chromosome pair is written in italics.
6. It would be appropriate and informative if you include the diplotype of the control population for comparative purposes in tables 7 and 8.
a) In table 7, the diplotype distribution in the control population is already shown in the bottom line.b) Table 8 is a summary of a logistic regression analysis conducted on the patient population, not including controls (for whom some of the variables analyzed are either not applicable or missing)
7. Can you provide further information on how PN down-regulates PPAR alpha expression and thus fatty acid beta-oxidation? Line 308.
As also reported in point 1 section a, we added more information and a new reference about the mechanisms of action of PN and the related involvement of PPAR alpha. From line 309 of the original manuscript (but actually moved together with the whole paragraph in the introduction section, as requested by reviewer 1) we reported:
In detail, PN could exert its protective effects against hypertriglyceridemia and liver steatosis through the interaction with the subtype α6β4 of the integrins family and the subsequent activation of an intracellular
pathway involving Rac1 and c-JUN. Those events would lead to the inhibition of the PPAR-α promoter, ROR α, and the downregulation of PPAR-α [12].
8. Line 355, you seem to have explained the difference observed in your study with Caucasian population and previous studies with Asian people that showed serum PN levels correlated with liver disease progression. Can you expand this discussion to include the significant differences in the genotypes of the Asian and Caucasian population used in these studies? Line 456-469.
To the best of our knowledge the correlation between periostin genetics and liver diseases is unprecedented in the literature (thus including any liver pathological condition, not just NAFLD): the evidence of periostin genetics is limited at present mainly to the cardiology and pulmonology fields, as already stated in the Introduction section. As a matter of fact, all the hepatological studies we mentioned in our paper after an extensive review of the literature deal only with the correlation between the levels of periostin (circulating or at liver cellular level) and the liver histological damage (in some cases) or the degree of ultrasound steatosis (in most cases). However, taking into account a possible explanation of the marked differences between our research and the three major human studies available in the literature which were all centred on Chinese subjects (J. Clin. Invest. 2014, 124, 3501–3513. Endocrine 2016, 51, 91–100. Sci. Rep. 2016, 6, 37886.), that is the lack of a significant association between periostin levels and the presence of NAFLD in our research as stated in line 355, we postulated that (beyond some relevant methodological differences, for instance the fact that Asian researches were conducted only on ultrasound-diagnosed NAFLDs, unlike ours and that of Polyzos et al. (Endocrine 2017, 56, 438–441) where the diagnosis was rigorously made with liver biopsy, please see lines 372-374 of the original manuscript) it is reasonable to assume that genetics may play an important role.
This hypothesis derives -in our opinion- from at least two considerations: a) the known different weight of genetics for what concerns NAFLD development in Asian versus Caucasian populations, as stated in lines 456-463 of the original manuscript; b) a significant known difference in the prevalence of the different periostin SNPs and haplotypes again between Asian and Caucasian populations (as already described in the text lines 448-445 for what concerns Japanese and European subjects); no information is available for what concerns POSTN diplotypes. Therefore, the evidence of genetics is quite strong, but only indirect.
Taking into account all these considerations, we decided to add the following statements to the original manuscript:
a) from line 469 we stressed the importance of POSTN diplotypes which showed some significance at the multivariate analysis of factors associated with high periostin (although not in the NAFLD group) and, more importantly, were associated with liver histological damage as stated in lines 273-275:
Moreover, our study evidenced, for the first time, significant differences in steatosis grading and fibrosis staging among different diplotypes, which would confirm a somehow role of the genetic pattern of periostin expression in the onset of liver disease and its severity. Hence, since up to now no information is available in the literature about this issue, our data -although needing to be validated in larger casuistries- would be of particular relevance.
b) Since our study often cited Chinese researches as above described, we added the frequencies of periostin minor alleles in East Asians, as can be checked from the database NIH/NCBI dbSNP Short Genetic Variations. The new paragraph starting from line 448 of the original manuscript is now:
As a matter of fact, the frequencies of the minor alleles reported by Kanemitsu, et al. were 0.136, 0.278, and 0.330, for rs1028728, rs3829365, and rs9603226, respectively [19]. These frequencies were reported to be similar in Chinese subjects, taking into account the evidence provided by the aforementioned studies even if not directly focused on population genetics [12-14]: 0.052, 0.320, and 0.278, respectively [44-46]. In our
population, the same frequencies were instead 0.242, 0.041, and 0.140, respectively, and comparable to what reported in the general European population (0.263, 0.070, and 0.106, respectively) [44–46]. The same considerations can be made for the haplotype frequencies identified respectively by the Japanese researchers and us: haplotype 1, 0.322 vs 0; haplotype 2, 0.278 vs 0.596; haplotype 3, 0.133 vs 0.003; haplotype 4, 0.267 vs 0.025; haplotype 5, 0 vs 0.137; haplotype 6, 0 vs 0.239 [19].
9. Line 395 to 421 excellent discussion. You should provide a summary paragraph on where PN studies should be directed further concerning its use as a marker of HCC or advanced liver disease in different ethnic populations.
A summary paragraph starting from line 487 of original manuscript has been added:
To better address this issue further studies could be organized in populations of HCC Caucasian and Asian patients stratified as concerning POSTN polymorphisms (haplotype), liver disease etiology and severity of the tumor disease. PN could also be quantified in the liver (from bioptic samples or resections) as protein expression and/or mRNA in order to better clarify its relations with the variables just mentioned. The comparison between tissue expression and plasma levels could also add information about the “dynamics” of PN release and its real role as a reliable circulating marker for HCC. Moreover, on the ground of the results of this study showing a correlation between plasma PN and alpha fetoprotein, it could be worth examining the correlations with other circulating HCC markers (including those being validated in liquid biopsies such as glypican-3 and glutamine synthetase) in order to strengthen the power of PN as a biomarker for this disease.
The clinical implications of the results obtained about plasma PN in HCC could be related to outcome parameters normally used by clinicians in common clinical practice (e.g. survival rate, event-free survival or reintervention rate) or to the results of different therapeutic approaches, which could involve PN itself. In this context, the role of PN could become of higher importance considering the existence of PN antagonists, aptamers, which are modified nucleic acids that specifically bind PN and inhibit its function. To date, PN antagonists have been investigated in breast and gastric cancer; however, a wider understanding of the role and mechanisms of PN in hepatic inflammation and fibrosis may render PN antagonists an innovative therapeutic approach for HCC and, maybe, also for NASH when the stigmata of the metabolic syndrome are prevalent due to PN known pathophysiological role [47].
10. English language and style are fine/minor spell check required.
English in the manuscript was thoroughly rechecked and edited for language and form.

Round 2
Reviewer 1 Report
The authors of the manuscript have adequately addressed my comments.